# Living with Family Is Directly Associated with Regular Dental Checkup and Indirectly Associated with Gingival Status among Japanese University Students: A 3-Year Cohort Study

**DOI:** 10.3390/ijerph18010324

**Published:** 2021-01-05

**Authors:** Momoko Nakahara, Daisuke Ekuni, Kota Kataoka, Aya Yokoi, Yoko Uchida-Fukuhara, Daiki Fukuhara, Terumasa Kobayashi, Naoki Toyama, Hikari Saho, Md Monirul Islam, Yoshiaki Iwasaki, Manabu Morita

**Affiliations:** 1Department of Preventive Dentistry, Okayama University Graduate School of Medicine, Dentistry and Pharmaceutical Sciences, Okayama 700-8558, Japan; pric37ll@s.okayama-u.ac.jp (M.N.); dekuni7@md.okayama-u.ac.jp (D.E.); de18017@s.okayama-u.ac.jp (K.K.); de421015@s.okayama-u.ac.jp (T.K.); de422026@s.okayama-u.ac.jp (H.S.); p3a99o50@s.okayama-u.ac.jp (M.M.I.); mmorita@md.okayama-u.ac.jp (M.M.); 2Department of Preventive Dentistry, Okayama University Hospital, Okayama 700-8558, Japan; yokoi-a1@cc.okayama-u.ac.jp (A.Y.); de20006@s.okayama-u.ac.jp (Y.U.-F.); de20041@s.okayama-u.ac.jp (D.F.); 3Department of Oral Morphology, Graduate School of Medicine, Dentistry and Pharmaceutical Sciences, Okayama University, Okayama 700-8558, Japan; 4Health Service Center, Okayama University, Okayama 700-8530, Japan; yiwasaki@okayama-u.ac.jp

**Keywords:** lifestyle, dental health behavior, oral health, oral hygiene, gingivitis, behavioral sciences

## Abstract

Although some studies showed that lifestyle was associated with oral health behavior, few studies investigated the association between household type and oral health behavior. The aim of this prospective cohort study was to investigate the association between household type, oral health behavior, and periodontal status among Japanese university students. Data were obtained from 377 students who received oral examinations and self-questionnaires in 2016 and 2019. We assessed periodontal status using the percentage of bleeding on probing (%BOP), probing pocket depth, oral hygiene status, oral health behaviors, and related factors. We used structural equation modeling to determine the association between household type, oral health behaviors, gingivitis, and periodontitis. At follow-up, 252 students did not live with their families. The mean ± standard deviation of %BOP was 35.5 ± 24.7 at baseline and 32.1 ± 25.3 at follow-up. In the final model, students living with their families were significantly more likely to receive regular dental checkup than those living alone. Regular checkup affected the decrease in calculus. The decrease in calculus affected the decrease in %BOP over 3 years. Living with family was directly associated with regular dental checkups and indirectly contributed to gingival status among Japanese university students.

## 1. Introduction

After university admission, university students obtain more freedom and independence. The beginning of university school life is often the first time that young people assume the responsibility to choose daily habits. Usually, university students in Japan are divided into two categories according to household type: Those who continue to live with their family and those who live far from their usual residence and their family. In Japan, about 60% of national and public university students live alone apart from their family [1]. Some students living apart from their families easily develop negative life habits [2]. The young adult period at the university may influence future lifestyle choices, health, and quality of life. Therefore, proper lifestyle choices for university students are important for good health later in life.

Previous studies [3,4,5,6] reported the association between living alone and lifestyle. Some cross-sectional and cohort studies reported that living alone was a risk factor of harmful lifestyle habits for university students, such as eating choices (food type) [3,4,5] and alcohol consumption [6]. Another cross-sectional study showed that university students who lived with their families had a greater chance of having good oral health-related dietary behaviors than those who lived alone [7]. In the field of dentistry, some cross-sectional and cohort studies mentioned that adults living alone were less likely to receive dental checkups [8,9]. However, few cohort studies investigated the association between living alone, oral health behavior, and oral diseases.

Among oral diseases, periodontal disease, including gingivitis and periodontitis, is one of the major diseases among young adults. It is an inflammatory disease of soft and/or hard tissues surrounding the tooth that is caused by the accumulation of bacterial biofilm (dental plaque) [10]. Periodontal disease is a risk factor for tooth loss [11]. Thus, prevention of periodontal disease, especially in the early stage or young adulthood, is very important.

Avoiding plaque accumulation with good oral hygiene behavior prevents periodontal disease. Some previous studies reported that oral health behaviors such as tooth brushing, dental floss use, and regular dental checkup affect oral hygiene and periodontal status in university students and adults [12,13,14].

Therefore, we hypothesized that university students living alone away from their families would have worse oral health behaviors and subsequently develop periodontal disease, compared with those living with families. The purpose of this cohort study was to analyze the association between household type (continuing to living with family or living alone far from family), oral health behavior, and periodontal status in university students.

## 2. Materials and Methods

### 2.1. Ethics Statement

The protocol of this study was approved by the Ethics Committee of Okayama University Graduate School of Medicine, Dentistry, and Pharmaceutical Sciences (No. 1060). All methods were performed in accordance with the Declaration of Helsinki. Informed consent was verbally obtained from each participant. Study reporting conformed to STROBE guidelines.

### 2.2. Study Population

We estimated sample size using G*Power and calculated the minimum sample size. We set an alpha of 0.05, power (1−β) of 0.80, and an effect size of 0.31 based on our preliminary research. The sample size was 264.

First-year students volunteered to receive oral examinations and answer self-reported questionnaires at the Health Service Center of Okayama University in April 2016 (baseline). Participants were recruited from all faculties (Faculties of Letters, Education, Law, Economics, Science, Pharmaceutical Sciences, Engineering, Environmental Science and Technology, Agriculture, Medicine, and Dentistry). After a 3-year follow-up, students volunteered to receive the oral examination and answer the questionnaire again (April 2019).

Inclusion criteria were students aged < 20 years and students who provided complete data at baseline. Exclusion criteria were students who had already lived apart from their family before university admission and students who provided incomplete data. Furthermore, we excluded students who had a history of smoking because smoking was expected to affect periodontal status [10]. We treated the data of students who did not receive the oral examination at follow-up as missing data. We widely promoted students to receive oral examinations to prevent selection bias due to loss of follow-up.

### 2.3. Self-Questionnaires

A self-administered questionnaire was delivered to each student before participating in dental examinations at baseline and follow-up. In addition to sex and age, the questionnaire included the following items. The students were asked about the following oral health behaviors: Daily frequency of tooth brushing (≥two times/≤one time); dental floss use (yes/no); receiving regular dental checkup (yes/no) [12,14,15]; and household type (living with family/living alone).

### 2.4. Oral Examination

Twelve dentists (D.E., T.Y., K.K., M.Y-T., A.T., A.Y., Y.U-F., D.F., T.K., N.T., K.F., and H.S.) examined oral status at baseline and follow-up. The following 10 teeth were selected for periodontal examination: Two molars in each posterior sextant and the upper right and lower left central incisors. The periodontal status was assessed using the Community Periodontal Index (CPI) [16] using a CPI probe (YDM, Tokyo, Japan) to evaluate 6 sites on each tooth (mesio-buccal, mid-buccal, disto-buccal, disto-lingual, mid-lingual, and mesio-lingual). Bleeding on probing (BOP) was an earlier and more sensitive indicator of inflammation than probing depth [17]. Therefore, in this study, the percentage of teeth exhibiting BOP (%BOP) was assessed among the 10 examined teeth using a CPI probe as an earlier sign of periodontal disease or gingivitis [12,14]. The level of dental plaque and calculus was assessed using the Debris Index-Simplified (DI-S) and Calculus Index-Simplified (CI-S) of Oral Hygiene Index-Simplified (OHI-S) [18]. Periodontal disease was defined using pocket scores of CPI criteria: Presence of probing pocket depth (PPD) ≥ 4 mm (pocket score = 1 or 2) at baseline and follow-up. After training the examiners, the CPI score in the 10 teeth was recorded and repeated within a 2-week interval in 2 volunteers. Intra- and inter-examiner reliabilities, evaluated by κ statistics, of CPI score were >0.8.

### 2.5. Statistical Analyses

Statistical analyses were conducted using SPSS version 22 (IBM, Tokyo, Japan). The normality of data was investigated by the Shapiro–Wilk test. We did not confirm the normal distribution of each value. The chi-squared test and Mann–Whitney *U* test were used to determine the significance of differences in sex, age, oral health behavior (tooth brushing frequency, dental floss use, and regular dental checkup), and oral status (DI-S, CI-S, %BOP and PPD). Values of *p* < 0.05 were considered significantly different.

Pathway analysis was performed to reveal the process from household type, oral health behavior, oral hygiene status, to periodontal status. We estimated an initial model (full model) with all the hypothesized pathways. Figure 1 shows an ideal model based on our hypothesis. Household type and oral health behavior indices were categorical indices. Changes in DI-S, CI-S, and %BOP during the 3-year study period were calculated by subtracting the baseline value from the follow-up value. These values were used as continuous variables. Change in PPD was divided into 3 groups according to the presence or absence of periodontal pockets of ≥ 4 mm at baseline and follow-up: (1) Pocket score = 0 at follow-up; (2) pocket score = 1 or 2 at baseline and follow-up; (3) pocket score = 0 at baseline, and 1 or 2 at follow-up, which was used as a categorical variable.

Mplus version 8.2 software (Muthén and Muthén, Los Angeles, CA, USA) was used for pathway analysis. Continuous and categorical variables were included in our data. Therefore, the pathway analysis was performed using weighted least-squares parameter estimates (WLSMV). WLSMV uses a diagonal weight matrix with robust standard errors and mean and variance adjusted chi-squared test statistics. The goodness of fit of the model was assessed using the comparative fit index (CFI), root mean square error of approximation (RMSEA), and the Tucker–Lewis coefficient (TLI) [12,19]. An RMSEA value < 0.05 suggested adequate fit, whereas CFI and TLI represented incremental fit; values > 0.95 indicated an adequate fit, whereas those > 0.90 were still acceptable [19,20]. Non-significant paths were removed in a step-by-step approach.

## 3. Results

Figure 2 shows a flowchart of participants included in this 3-year cohort study from baseline to follow-up. At baseline, 2174 students received voluntary oral examinations and answered self-reported questionnaires. We selected 2026 students who met the inclusion criteria. At follow-up, we excluded 1649 students who did not undergo an oral examination and matched exclusion criteria. Finally, we analyzed 377 students (follow-up rate, 18.6%).

Table 1 shows the characteristics of the participants. This study analyzed data from 377 participants (187 males and 190 females; aged 18–19 years at baseline). At follow-up, 252 students (66.8%) did not live with their families (133 males, 52.8%; 119 females, 47.2%). Mean %BOP value was 35.5 ± 24.7 (mean ± standard deviation) at baseline and 32.1 ± 25.3% at follow-up.

Table 2 shows the results of self-questionnaires and the oral examination according to the two household types at follow-up. At baseline and follow-up, household type was not associated with sex, age, daily frequency of tooth brushing, or dental floss use (*p* > 0.05). At follow-up, participants who were living with their families reported significantly greater regular dental checkups than participants who were not (*p* = 0.004); no such difference was observed at baseline. At baseline and follow-up, there were no significant differences in DI-S, CI-S, OHI-S, %BOP, and PPD between students living with family and students living alone.

The final model (Figure 3) was estimated with only statistically significant paths retained. Figure 3 shows parameter estimates for the final structural model. The final model exhibited good fit (χ^2^ = 2.366; df = 3; *p* = 0.500; CFI = 1.000; TLI = 1.032; RMSEA = 0.000 [0.000–0.079]). The model indicated that (1) students who lived with their families were significantly more likely to receive regular dental checkups than those who lived alone; (2) regular checkup affected the decrease in calculus, and (3) the decrease in calculus affected the decrease in %BOP over the 3-year study period. 

## 4. Discussion

To the best of our knowledge, this study is the first prospective cohort study to investigate the association between household type, oral health behavior, and periodontal status in young adults using the pathway analysis. Pathway analysis enables variables to act both as independent and dependent variables and explore the complex causal relationship involved in disease processes [21]. We showed that students who lived with their families were significantly more likely to receive regular dental checkups than those who lived alone. Receiving regular dental checkups directly affected the decrease in calculus. The decrease in calculus, in turn, directly affected the decrease in %BOP (gingival status) over the 3-year study period. After all, the household type was indirectly associated with %BOP through regular dental checkups and calculus (CI-S).

Living with families was directly associated with a regular dental checkup. Oral health behaviors of students living with their families may be influenced by familial oral health behaviors, and students may be encouraged to receive regular dental checkups. A previous study reported that parents’ dental visits within the previous year significantly affected regular dental checkups for secondary school students [22]. Moreover, adults living with a partner or spouse enhanced the chance of receiving dental checkups [8,9].

Living away from family may have a negative effect on receiving regular dental checkups through an indirect factor; the financial difficulties of students living alone. Living alone away from family may increase financial burden because of increased costs related to living alone compared to living with others [23]. Socioeconomic status was associated with regular dental checkups, although the participants in these reports were not university students [13,24,25].

Receiving regular dental checkups affected the decrease in calculus in this study. Our result supports the findings of previous studies indicating a positive effect of a dental checkup on oral hygiene status [12,13,14]. According to previous studies, tooth brushing, dental floss use, and receiving regular dental checkups were significantly associated with good oral hygiene status in Japanese university students [12,14]. Oral health education from dental clinics showed to be effective in improving oral health behavior, knowledge, and oral hygiene status of adolescents [12,14,26]. Dentists play an important role in the prevention of oral disease of general people [27,28]. Continuing oral health education effectively maintained oral health behavior, including regular dental checkups [29,30,31]. Taken together, it is quite reasonable that receiving regular dental checkups affected the decrease in calculus.

In the final model, the decrease in calculus affected the decrease in %BOP. Our result partially supports those of previous studies [12,14], indicating a significant association between oral hygiene status and %BOP in Japanese university students. By contrast, the decrease in dental plaque did not affect the decrease in %BOP, which is inconsistent with the results of previous studies [12,14]. Gingivitis is caused by the continuous accumulation of dental plaque [10], but DI-S is closely related to oral hygiene status immediately before data acquisition. CI-S may be more effective as an indicator of current gingivitis rather than DI-S.

The results of this study might be clinically relevant. When clinicians encounter younger patients living away from their families, increased efforts may be needed to prevent periodontal disease. This work was an observational study. Further studies are, therefore, needed to clarify whether clinical interventions can help prevent periodontal disease in young people for the young living alone.

In this final model, there was no relationship between living alone and tooth brushing or dental floss use. Although no such significant associations have been observed in adults or university students in previous studies, oral self-care such as tooth brushing and dental floss use may not be affected by familial behaviors compared to the effect on visiting the dental office. Furthermore, this result may be influenced by floor and ceiling effects because a very low percentage of students used floss, and a very high percentage of students brushed their teeth twice or more per day.

The change in PPD was excluded in the final model. BOP is an earlier and more sensitive indicator of inflammation than probing depth [17]. Therefore, in this study, the final model may include %BOP but not PPD.

In this study, university students’ oral health behavior tended to be better than in the previous prospective 3-year cohort studies in university students [32,33]. In the present study, 30.2% of students answered received regular dental checkups at follow-up. On the other hand, the previous 3-year cohort studies showed a lower prevalence (10.5–14.0%). The percentage of students who answered that they brushed their teeth more than once (87.5 %) was slightly higher than the previous cohort studies (75.0–84.7%). The percentage of students who answered they used dental floss (16.7%) was also slightly higher than the previous cohort studies (10.7–15.4%) [32,33].

There are some limitations associated with our study. First, we did not investigate whether students’ families had a dental checkup and the recall interval for a dental checkup. Previous studies investigated the association between secondary school students’ regular dental checkups and parents’ most recent dental visits [22]. Second, we did not consider factors that may be associated with periodontal diseases, such as students’ faculties [34], students’ and their families’ education level [35,36], socioeconomic status [37,38], psychosocial factors [39], and social capital [15], in this study. Future studies are needed to reveal these effects. Third, there may have been a selection bias, given the low follow-up rate. In this study, analyzed students (n = 377) comprised 18.6% of all eligible students (n = 2026). The response rate in this study was within the previous prospective 3-year cohort studies in university students (13.4–25.7%) [32,33,40,41]. No significant differences were seen in %BOP and sex ratios between analyzed and non-analyzed students (377 vs. 1649 students; Mann–Whitney *U* test: *p* > 0.05, chi-squared test: *p* > 0.05). Any effects of a selection bias would have, therefore, been negligible. Forth, we examined only 10 teeth in the oral examination, which might have led to under- or overestimation. We may have a bias that a full examination of all teeth about dental plaque, calculus, PPD, and BOP can be avoided. Finally, all participants were recruited from Okayama University students. This may limit the ability to extrapolate these findings to the general young population.

## 5. Conclusions

Living with family was directly associated with regular dental checkups and indirectly associated with gingival status among Japanese university students. When clinicians encounter younger patients living away from their families, increased efforts may be needed to prevent periodontal disease.

## Figures and Tables

**Figure 1 ijerph-18-00324-f001:**
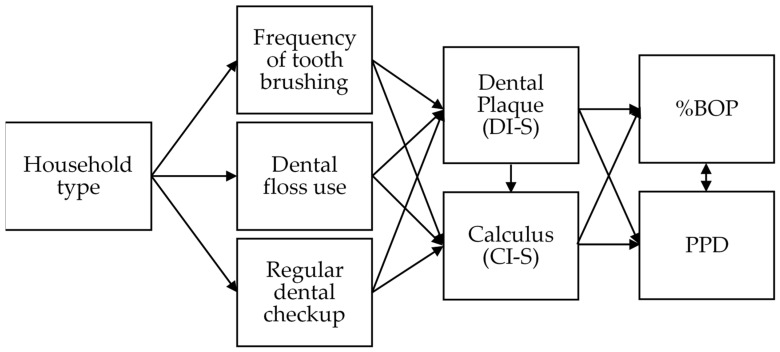
Pathway based on hypothesis showing the association between household type, oral health behavior, oral hygiene status; Debris Index-Simplified (DI-S) and Calculus Index-Simplified (CI-S), percentage of bleeding on probing (%BOP), and probing pocket depth (PPD).

**Figure 2 ijerph-18-00324-f002:**
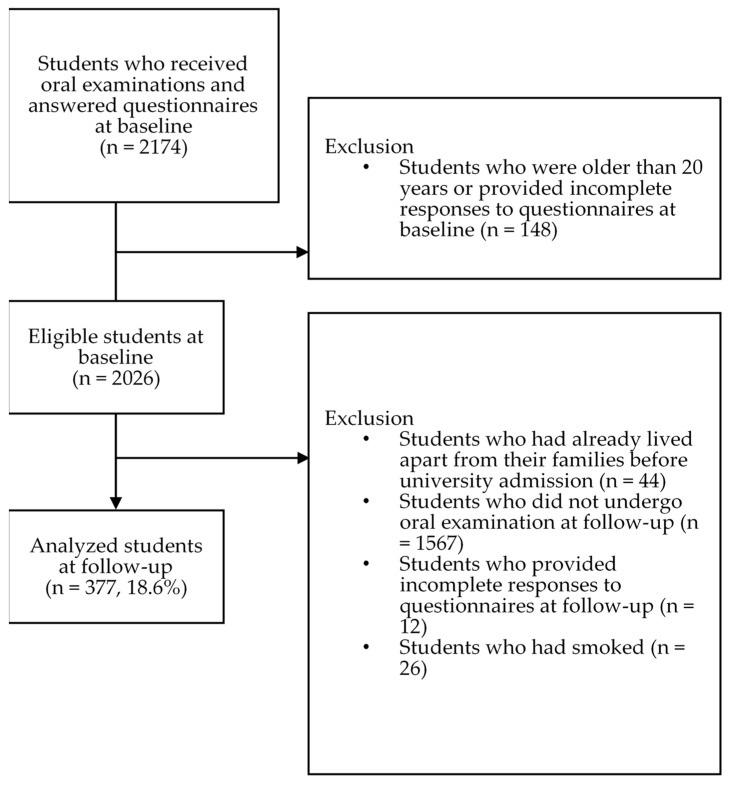
Flowchart of students included in analyses.

**Figure 3 ijerph-18-00324-f003:**
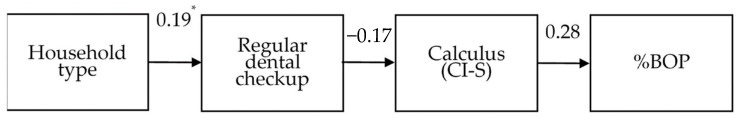
Pathway based on structural equation modeling showing the association between household type, oral health behavior, oral hygiene status, and the percentage of bleeding on probing (%BOP). * Values above single-headed arrows indicate standardized coefficients.

**Table 1 ijerph-18-00324-t001:** Participant characteristics (n = 377).

Parameter	Baseline	Follow-Up
Sex		
Male	187 (49.6) *	−
Female	190 (50.4)	−
Age (year)		
18	325 (86.2)	−
19	52 (13.8)	−
Tooth brushing (daily frequency)		
≤One time	40 (10.6)	47 (12.5)
≥Two times	337 (89.4)	330 (87.5)
Dental floss use		
No	304 (80.6)	314 (83.3)
Yes	73 (19.4)	63 (16.7)
Regular dental checkup		
No	258 (68.4)	263 (69.8)
Yes	119 (31.6)	114 (30.2)
OHI-S	0.50 (0.17, 0.83) ^†^	0.33 (0.00, 0.67)
DI-S (dental plaque)	0.33 (0.00, 0.67)	0.17 (0.00, 0.67)
CI-S (calculus)	0.00 (0.00, 0.33)	0.00 (0.00, 0.08)
Percentage of teeth exhibiting BOP (%BOP)	35.5 ± 24.7	32.1 ± 25.3
Probing pocket depth (PPD)		
≤3 mm	284 (75.3)	206 (54.6)
≥4 mm	93 (24.7)	171 (45.4)
Household type		
Living alone	0 (0.0)	252 (66.8)
Living with family	377 (100.0)	125 (33.2)

DI-S; Debris Index-Simplified, CI-S; Calculus Index-Simplified, OHI-S; Oral Hygiene Index-Simplified. * Number (percentage), ^†^ median (25th percentile, 75th percentile).

**Table 2 ijerph-18-00324-t002:** Differences in sex, age, oral health behavior, and oral status by household type at baseline and follow-up.

Parameter	Baseline			Follow-Up		
	Living Alone at Follow-Up (n = 252)	Living with Family at Follow-Up (n = 125)	*p* ^‡^	Living Alone at Follow-Up (n = 252)	Living with Family at Follow-Up (n = 125)	*p* ^‡^
Sex						
Male	133 (52.8) *	54 (43.2)	0.080	-	-	-
Female	119 (47.2)	71 (56.8)		-	-	
Age (year)						
18	212 (84.1)	113 (90.4)	0.096	-	-	-
19	40 (15.9)	12 (9.6)		-	-	
Tooth brushing (daily frequency)						
≤One time	23 (9.1)	17 (13.6)	0.184	33 (13.1)	14 (11.2)	0.600
≥Two times	229 (90.9)	108 (86.4)		219 (86.9)	111 (88.8)	
Dental floss use						
No	205 (81.3)	99 (79.2)	0.619	212 (84.1)	102 (81.6)	0.536
Yes	47 (18.7)	26 (20.8)		40 (15.9)	23 (18.4)	
Regular dental checkup						
No	175 (69.4)	83 (66.4)	0.549	188 (74.6)	75 (60.0)	0.004
Yes	77 (30.6)	42 (33.6)		64 (25.4)	50 (40.0)	
OHI-S	0.500 (0, 0.830) ^†^	0.330 (0.170, 0.830)	0.611	0.333 (0, 0.667)	0.333 (0, 0.750)	0.941
DI-S	0.330 (0, 0.670)	0.330 (0.17, 0.670)	0.621	0.167 (0, 0.667)	0.167 (0, 0.667)	0.855
CI-S	0 (0, 0.330)	0 (0, 0.330)	0.665	0 (0, 0)	0 (0, 0.167)	0.455
Percentage of teeth exhibiting BOP (%BOP)	30.0 (10.0, 50.0)	40.0 (20.0, 50.0)	0.580	30.0 (10.0, 50.0)	30.0 (10.0, 50.0)	0.941
Probing pocket depth (PPD)						
≤3 mm	186 (73.8)	98 (78.4)	0.330	144 (57.1)	62 (49.6)	0.166
≥4 mm	66 (26.2)	27 (21.6)		108 (42.9)	63 (50.4)	

DI-S; Debris Index-Simplified, CI-S; Calculus Index-Simplified, OHI-S; Oral Hygiene Index-Simplified. * Number (percentage), ^†^ median (25th percentile, 75th percentile), ^‡^ Chi-squared test or Mann–Whitney *U* test.

## Data Availability

The data presented in this study are available on request from the corresponding author. The data are not publicly available due to ethical issues.

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
