# Peer review of "Living with Family Is Directly Associated with Regular Dental Checkup and Indirectly Associated with Gingival Status among Japanese University Students: A 3-Year Cohort Study"

_ijerph, 2021, doi:10.3390/ijerph18010324_

Round 1
Reviewer 1 Report
Thank you for submitting this manuscript.
Your introduction provides adequate background and justification for your study. You clearly state your purpose.
The methods are clearly stated and you have intra- and interrater reliability scores for the 12 dentists. My suggestion would be to explain the calibration process.
Results:
- How does the recall rate of 18.6% compare to similar studies. This would be helpful in gaging if you had greater amount or lesser amount of people drop out compared to similar studies.
- Line 162-164 you mention that at baseline and follow-up there was no sig. difference in DI-S, CI-S, OHI-S, %BOP, and PPD between students living with family and students living along. However in your final model described in lines 171-176 you state there is a difference between these groups. I think it would be helpful to explain how you came to this with the help of figure 1 and 3.
Discussion
- Line 185-186 "Receiving regular dental checkup directly affected the decrease in calculus." this seems at odds with the statement in line 171-176. please explain this further.
- It would be interesting to see this study conducted across multiple Universities and also see if there is a difference with education level (undergrad vs. grad) or education level of parents.
- In selecting 10 teeth to examine you may have created a bias that a full exam of all teeth would have avoided. I am glad you commented on this potential limitation.
Author Response
Response to Reviewer 1 Comments
Thank you for submitting this manuscript.
Your introduction provides adequate background and justification for your study. You clearly state your purpose.
Our response: Thank you for your comments.
The methods are clearly stated and you have intra- and interrater reliability scores for the 12 dentists. My suggestion would be to explain the calibration process.
Our response: Thank you for your comments. We have added the comments in the materials and methods following the reviewer’s suggestion (L111-112: Materials and Methods: Oral examination).
Results:
- How does the recall rate of 18.6% compare to similar studies. This would be helpful in gaging if you had greater amount or lesser amount of people drop out compared to similar studies.
Our response: Thank you for your comments. We have added the comments in the discussion following the reviewer’s suggestion (L249-250: Discussion).
- Line 162-164 you mention that at baseline and follow-up there was no sig. difference in DI-S, CI-S, OHI-S, %BOP, and PPD between students living with family and students living along. However in your final model described in lines 171-176 you state there is a difference between these groups. I think it would be helpful to explain how you came to this with the help of figure 1 and 3.
Our response: Thank you for your comments. In a multivariate analysis, many researchers prefer to use the logistic regression or multiple linear regression analysis. These analyses are set on only one dependent variable. In other words, these analyses enable one to examine direct effects from independent variables to dependent variable, not indirect effects. In addition, they cannot reveal complex and diverse relationships between independent variables and dependent variables. On the other hand, pathway analysis enables variables to act both as independent and dependent, and explore the complex causal relationship involved in disease processes.
Thus, we used the pathway analysis in this study. We showed that students who lived with their families were significantly more likely to receive regular dental checkup than those who lived alone. Receiving regular dental checkup directly affected the decrease in calculus. The decrease in calculus, in turn, directly affected the decrease in %BOP (gingival status) over the 3-year study period. After all, household type was indirectly associated with %BOP through regular dental checkup and calculus (CI-S). Then, the finding was different with the results of direct analysis that there were no significant differences in DI-S, CI-S, OHI-S, %BOP, and PPD between students living with family and students living along.
We have added the comments in the discussion following the reviewer’s suggestion (L186-187, 191-192: Discussion).
Discussion
- Line 185-186 "Receiving regular dental checkup directly affected the decrease in calculus." this seems at odds with the statement in line 171-176. please explain this further.
Our response: Thank you for your comments. We have changed standardized coefficients between regular dental checkup and calculus (CI-S) in the final model to -0.17 (Figure 3) to understand easily. We have changed definition of categorical indices as follows: household type (living with family; 1/living alone; 0): daily frequency of tooth brushing (≥ two times; 1/≤ one time; 0); dental floss use (yes; 1/no; 0); receiving regular dental checkup (yes; 1/no; 0).
- It would be interesting to see this study conducted across multiple Universities and also see if there is a difference with education level (undergrad vs. grad) or education level of parents.
Our response: Thank you for your comments. The conclusion might be more reliable if similar results are observed across multiple Universities in our future study. As the reviewer pointed out, we should pay attention to confounders that may affect periodontal status. We did not investigate education level of parents, so we have added the comments in the discussion following the reviewer’s suggestion (L245-246: Discussion).
- In selecting 10 teeth to examine you may have created a bias that a full exam of all teeth would have avoided. I am glad you commented on this potential limitation.
Our response: Thank you for your comments. We have added the comments in the discussion following the reviewer’s suggestion (L254-255: Discussion).

Reviewer 2 Report
The study aims to investigator amongst university students the influence of family environment on oral health behavour which in turn may influence oral health status. The use of SEM in analysis is novel, a reasonable approach and appropriate. The limitations of the study was also discussed
While the approach &methodology is sound, the following queries are raised :
1 for assessment of oral hygiene, why was the debris index used? the index is rather crude in assessing oral hygiene in today's context, unless the authors have modified the index, please clarify
2 The final sample size was rather small to derive significant benefits of living at home in terms of oral health behaviour.
3 it would be useful to provide some background or available knowledge or data ( if any) of oral health behaviour of similar cohort of university students
Author Response
Response to Reviewer 2 Comments
The study aims to investigator amongst university students the influence of family environment on oral health behavour which in turn may influence oral health status. The use of SEM in analysis is novel, a reasonable approach and appropriate. The limitations of the study was also discussed
While the approach &methodology is sound, the following queries are raised:
Our response: Thank you for your comments. We have revised the manuscript based on reviewers’ comments as below.
- for assessment of oral hygiene, why was the debris index used? the index is rather crude in assessing oral hygiene in today's context, unless the authors have modified the index, please clarify
Our response: Thank you for your comments. We used Oral Hygiene Index-Simplified (OHI-S), which evaluates both calculus and plaques in a short period of time and is useful in epidemiological studies. However, we may have a bias. Thus, we have added the comments about limitation (L254-255: Discussion).
- The final sample size was rather small to derive significant benefits of living at home in terms of oral health behaviour.
Our response: Thank you for your comments. In this study, analyzed students (n = 377) comprised 18.6% of all eligible students (n = 2,026). However, the sample seize was enough based on the estimation. Furthermore, no significant differences were seen in %BOP and sex ratios between analyzed and non-analyzed students (377 vs. 1,649 students; Mann–Whitney U test: P > 0.05, chi-squared test: P > 0.05). Any effects of a selection bias would have therefore been negligible (L250-253: Discussion). It is necessary to increase the number of samples for further research in the future.
- it would be useful to provide some background or available knowledge or data (if any) of oral health behaviour of similar cohort of university students
Our response: Thank you for your comments. We have added the comments in the discussion following the reviewer’s suggestion (L234-240: Discussion).

Round 2
Reviewer 1 Report
Thank you for sending your edits and responses. I have no further comments.